# Human G-MDSCs are neutrophils at distinct maturation stages promoting tumor growth in breast cancer

Meliha Mehmeti-Ajradini[1], Caroline Bergenfelz[2,*], Anna-Maria Larsson[3,4,*], Robert Carlsson[5], Kristian Riesbeck[6] , Jonas Ahl[7], Helena Janols[7], Marlene Wullt[7], Anders Bredberg[6], Eva Källberg[1], Frida Björk Gunnarsdottir[1], Camilla Rydberg Millrud[1], Lisa Rydén[3,8] , Gesine Paul[5], Niklas Loman[3,4], Jörgen Adolfsson[9], Ana Carneiro[3,4] , Karin Jirström[10], Fredrika Killander[3,4,†], Daniel Bexell[11,†], Karin Leandersson[1]

**Myeloid-derived suppressor cells (MDSCs) are known to contribute to immune evasion in cancer. However, the function of the human granulocytic (G)-MDSC subset during tumor progression is largely unknown, and there are no established markers for their identification in human tumor specimens. Using gene expression profiling, mass cytometry, and tumor microarrays, we here demonstrate that human G-MDSCs occur as neutrophils at distinct maturation stages, with a disease-specific profile. G-MDSCs derived from patients with metastatic breast cancer and malignant melanoma display a unique immature neutrophil profile, that is more similar to healthy donor neutrophils than to G-MDSCs from sepsis patients. Finally, we show that primary G-MDSCs from metastatic breast cancer patients co-transplanted with breast cancer cells, promote tumor growth, and affect vessel formation, leading to myeloid immune cell exclusion. Our findings reveal a role for human G-MDSC in tumor progression and have clinical implications also for targeted immunotherapy.**

## Introduction

Avoidance of immune surveillance enables tumor development and is one of the hallmarks of cancer (Hanahan & Weinberg, 2011). One mechanism exploited by cancer to evade immune destruction is the accumulation of immunosuppressive myeloid-derived suppressor cells (MDSCs) (Swann & Smyth, 2007). These cells are immature immunosuppressive myeloid cells that are generated in most patients with advanced cancer, causing T-cell suppression by mediators, for example, reactive oxygen species (ROS), iNOS, or arginase I. MDSCs might constitute a good target for anti-cancer therapies, but major challenges in defining their nature in humans have until now prevented specific targeting. Furthermore, although MDSCs have been predominantly described in cancer, they are also implicated in other pathological conditions, for example sepsis (Janols et al, 2014; Kontaki et al, 2017). However, the relationship between MDSCs in different diseases, both regarding their identity and function, remains obscure.

The key characteristic of all MDSCs is their immunosuppressive function (Bronte et al, 2016). Two main classes of MDSCs are currently recognized: monocytic MDSCs (Mo-MDSCs) and granulocytic MDSCs (G-MDSCs), also designated PMN-MDSCs. Generally, Mo-MDSCs are derived from the monocytic cell lineage, whereas G-MDSCs are derived from the granulocytic myeloid cell counterpart. In humans, the following cell surface phenotypes of Mo-MDSCs and G-MDSCs are recognized: $CD11b^+CD14^+CD33^+HLA-DR^{low/-}Co-receptor^{low/-}$ and $CD15^+CD33^+CD11b^+CD66b^+CD14-HLA-DR^{low/-}$, respectively (Elliott et al, 2017; Gabrilovich, 2017). G-MDSCs are further characterized by low density in Ficoll gradient centrifugations (low density granulocytes; LDGs) and a granulocytic scatter profile on flow cytometry (FSC/SSC) (Elliott et al, 2017; Gabrilovich, 2017). G-MDSCs are of special interest not only because of their masked relatedness to neutrophils that are also increased in cancer patients and associated with worse prognosis but also because of their unknown functions other than immunosuppression in humans.

The generation and identity of G-MDSCs in humans remains a controversial case for debate but is vital for specific therapeutic targeting of G-MDSCs. A heterogeneous cell morphology, ranging

[1]Department of Translational Medicine, Cancer Immunology, Lund University, Malmö, Sweden    [2]Division of Experimental Infection Medicine, Department of Translational Medicine, Lund University, Malmö, Sweden    [3]Division of Oncology, Department of Clinical Sciences, Lund University, Skåne University Hospital, Lund, Sweden [4]Department of Hematology, Oncology and Radiation Physics, Skåne University Hospital, Lund, Sweden    [5]Translational Neurology, Department of Clinical Sciences and Wallenberg Centrum for Molecular Medicine, Lund University, Lund, Sweden    [6]Department of Translational Medicine, Clinical Microbiology, Lund University, Malmö, Sweden    [7]Department of Infectious Diseases, Department of Translational Medicine, Lund University, Skåne University Hospital, Malmö, Sweden    [8]Department of Surgery and Gastroenterology, Skåne University Hospital, Lund, Sweden    [9]Science for Life Laboratory Node at Linköping's University, Linköping, Sweden    [10]Department of Clinical Sciences, Oncology and Therapeutic Pathology, Lund University, Lund, Sweden    [11]Division of Translational Cancer Research, Department of Laboratory Medicine, Lund University, Lund, Sweden

Correspondence: Karin.Leandersson@med.lu.se
*Caroline Bergenfelz and Anna-Maria Larsson contributed equally to this work
†Fredrika Killander and Daniel Bexell contributed equally to this work

from blast-like (myelocyte) to PMN nucleus (Fig 1A), has been described for G-MDSCs in both cancer and sepsis patients (Janols et al, 2014; Sagiv et al, 2015; Millrud et al, 2017; Mackey et al, 2019). Cells identical to G-MDSCs are also present in autoimmune diseases, but here have pro-inflammatory functions (Silvestre-Roig et al, 2016). Different theories have been proposed to explain the nature of G-MDSCs in humans. According to one theory (Pillay et al, 2013), G-MDSC formation is driven by various tumor-derived cytokines and growth factors that induce aberrant emergency myelopoiesis, resulting in an increased proportion of immature neutrophils, that is, G-MDSCs. Another theory draws on their similar morphological and phenotypic characteristics to propose that G-MDSCs are, in fact, a heterogeneous subset of alternatively activated neutrophils (Rodriguez et al, 2009; Pillay et al, 2012, 2013; Condamine et al, 2016; Millrud et al, 2017). This notion comes from mouse studies where tumor infiltrating neutrophils (TANs) are regarded as G-MDSCs (Ly6G$^+$) (Lecot et al, 2019). Indeed, both activated, degranulated neutrophils (Sippel et al, 2011) and myelocytes (Sagiv et al, 2015; Mackey et al, 2019) are detected in the low-density mononuclear cell fraction of Ficoll density gradients from cancer patients containing LDGs, a hallmark of G-MDSCs. Yet another theory explains G-MDSC origin in terms of cell plasticity, that is, they may be activated neutrophils but with a seemingly immature surface phenotype (Mackey et al, 2019), or may even be cells of fibrocyte origin (Zhang et al, 2013). Markers such as Lox-1 (*OLR1*), *ARG1*, *MMP8/9*, and *IDO1* have been proposed in the search for the bona fide G-MDSC (Condamine et al, 2016; Elliott et al, 2017; Gabrilovich, 2017). Nevertheless, the generation and identity of G-MDSCs in humans remains a matter of debate because the established human G-MDSC markers cannot be used to discriminate between immature, mature, or activated neutrophils (Fig 1A) (Bergenfelz & Leandersson, 2020). The fact that the proteomes of immature and mature neutrophils differ vastly may be important when designing therapies that target G-MDSCs (Mackey et al, 2019).

Although the immunosuppressive role of G-MDSCs is relatively well established (Elliott et al, 2017), much less is known about the tumor-promoting activity of these cells in humans. Most studies regarding the tumor-promoting function of G-MDSCs have historically focused on murine tumor models. In mice, G-MDSCs in addition to immunosuppressive roles, promote tumor cell proliferation, angiogenesis, and metastatic growth (Yang et al, 2004; Ouzounova et al, 2017; Zhou et al, 2018), which is in contrast to Mo-MDSCs (Ouzounova et al, 2017). However, conflicting data exist on a potential tumor-promoting role of G-MDSCs (Bronte et al, 2016; Eruslanov et al, 2017; Zhou et al, 2018). The tumor-promoting function of human G-MDSCs per se has never been investigated in vivo, primarily because of the difficulties in investigating patient-derived primary cells with short half-life.

Considering the potentially therapeutic importance of G-MDSCs in cancer and other malignancies, we aimed to determine the nature of primary human G-MDSCs from various patient cohorts, their origin, and function in vivo. Here, we show that the G-MDSC population in peripheral blood in MBC patients is increased compared with healthy donors, but that G-MDSC levels do not correlate with disease severity. The primary human G-MDSCs in cancer patients rather represent neutrophils at different maturation stages, correlating with peripheral neutrophil counts, thus

supporting the aberrant emergency myelopoiesis model of G-MDSC generation. Furthermore, we demonstrate that in a mouse xenograft model, primary cancer patient–derived G-MDSCs are capable of affecting tumor growth, vessel formation, and myeloid immune cell exclusion in vivo by a mechanism that does not appear to involve ROS. Finally, based on the analysis of a breast cancer (BC) patient cohort, we demonstrate that the presence of immature neutrophil G-MDSCs has a stronger prognostic value for cancer recurrence than cells representing activated neutrophils. In summary, we provide unique evidence that human G-MDSCs represent neutrophils at a range of maturation stages with tumor promoting capacities, indicating that the neutrophil lineage as such could be a therapeutic target.

## Results

### G-MDSCs are increased in patients with MBC and display heterogeneous morphology

We first asked whether G-MDSC levels change in individuals with different malignancies compared with healthy individuals. Accordingly, we investigated the presence of G-MDSCs in MBC patients before the start of systemic therapy, in patients with Gram-positive sepsis (as a positive control with a different causative disease [Janols et al, 2014]), and in healthy controls (Fig 1B). We determined the proportion (%) of G-MDSCs in the Ficoll-enriched PBMC fraction using flow cytometry and specific antibodies against CD15, CD33, CD11b, CD11c, and CD14, and also anti-CD64 antibodies for cell sorting (Fig 1C; representative dot plots and gating strategy is shown in Figs 1C and S1A, green boxes). The proportion of G-MDSCs was significantly increased in MBC patients compared with healthy donors (Fig 1B). High G-MDSC levels were seen in approximately one-third of MBC patients analyzed. Furthermore, G-MDSC levels in MBC patients were similar to those in Gram-positive sepsis patients (Fig 1B). This indicated that G-MDSC levels are elevated by various diseases.

We next performed an extensive phenotypic analysis to better characterize this cell population in MBC patients. A subpopulation of cells with a surface phenotype similar to that of G-MDSCs, termed fibrocytes, has been previously described in gastric cancer (Terai et al, 2015). These bone marrow–derived cells, expressing CD34, CD45, collagen I/III, and $\alpha$-smooth muscle actin ($\alpha$-SMA), can differentiate into fibroblasts and myofibroblasts and promote wound healing (Terai et al, 2015). We therefore also used markers such as HLA-DR, CD127, collagen I, $\alpha$-SMA, CD123, CD34, and CD66b (Fig S1A, pink boxes). G-MDSCs had a SSC$^{high}$/FSC$^{low/int}$ scatter profile (LDGs) and the phenotype CD15$^+$CD33$^+$CD11b$^+$CD11c$^+$CD66b$^+$CD64$^{-/low}$CD14$^{-/low}$HLA-DR$^{-/low}$CD127$^{int}$ CD123$^-$CD90$^-$CD34$^{-/+}$ColI$^-\alpha$-SMA$^-$ (Figs 1C and S1A, green and pink boxes). Monocytes from the same sample displayed a CD33$^+$CD11b$^+$CD11c$^+$CD15$^-$CD14$^{+/high}$CD64$^{+/high}$ surface profile (Fig 1C, orange boxes). We further set out to investigate the morphology and phenotype of G-MDSCs from MBC patients in more detail. Analysis of cytospin material of sorted G-MDSCs revealed that G-MDSCs morphologically comprised a heterogeneous cell population containing blast-like cells (probably representing myelocytes, <5%; orange arrows in Fig 1D), banded neutrophils (purple arrow), and, predominantly,

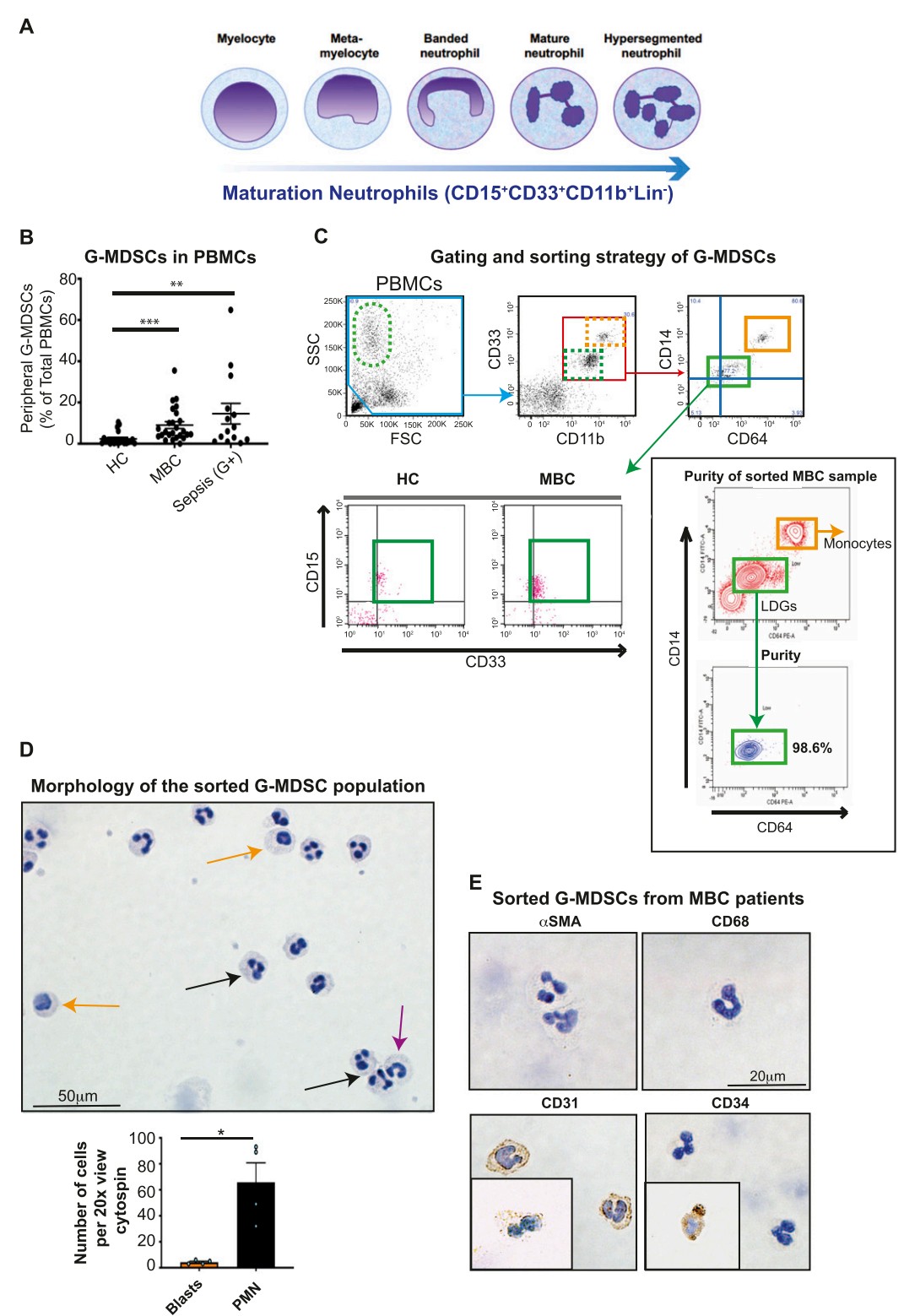

**Figure 1. The proportion of granulocytic myeloid-derived suppressor cells (G-MDSCs) is increased in patients with metastatic breast cancer (MBC).**
**(A)** An overview of neutrophil maturation in humans, with all cell stages having the CD15+CD33+CD11b+Lin− surface phenotype. **(B, C)** Flow cytometry analysis of freshly isolated PBMCs from healthy controls (HC), patients with MBC, and patients with Gram-positive sepsis. **(B, C)** Frequency of G-MDSCs (green box in C) among PBMCs (blue box in C) isolated from HC (*N* = 21), MBC (*N* = 25), and patients with Gram-positive sepsis (*N* = 14). Error bars indicate SEM. **P < 0.01, ***P < 0.001. ANOVA with multiple comparisons Kruskal–Wallis test. **(C)** Dot plots representing the gating and sorting strategy of G-MDSCs (green box) and monocytes (orange box) of PBMCs, with purity after sorting indicated. **(D)** G-MDSCs from MBC patients are a heterogeneous cell population. Cytospin fractions of sorted G-MDSCs from MBC patients were analyzed by HE

mature neutrophils with PMN cell morphology (>95%; black arrows in Fig 1D), similar to what we have previously shown in sepsis patients (Janols et al, 2014). We analyzed the sorted G-MDSCs by immunohistochemistry (IHC) (Fig 1E). The cells did not express the pan-macrophage marker CD68 or fibrocyte marker α-SMA, although they showed a heterogeneous expression of the endothelial cell marker CD31. A minor fraction of cells expressed the hematopoietic stem cell marker CD34. This was in agreement with the flow cytometry findings (Fig S1A) and, together with the flow cytometry data, ruled out the occurrence of fibrocytes or mesenchymal stem cells within the G-MDSC population.

We confirmed the immunosuppressive capacity and mechanism of sorted G-MDSCs using allogeneic T-cell suppression assays and ROS-specific inhibition assays using catalase (Fig S1B and C). Collectively, these experiments confirmed the unique phenotype of G-MDSCs from cancer patients, in addition to that the cells exert immunosuppressive effects ex vivo via ROS generation.

### G-MDSC levels do not correlate with disease severity in MBC patients

We recently showed that an increased proportion of Mo-MDSCs in a similar patient group of MBC patients correlates with disease severity and progression (Bergenfelz et al, 2015b). This led us to investigate whether the G-MDSC levels in MBC patients are associated with clinical parameters (Table 1). In the analysis, we defined "high" G-MDSC levels as those higher than the highest G-MDSC levels observed in healthy donors (>10.5%). Interestingly, the increased G-MDSC levels were not correlated with any clinicopathological variables or prognostic factors except for the peripheral neutrophil count ($P < 0.05$) (Table 1). This suggests that G-MDSC levels are induced in a fraction of patients with disseminated disease, with an already high peripheral neutrophil count. However, in contrast with Mo-MDSC levels (Bergenfelz et al, 2015b), the G-MDSC levels did not correlate with other clinicopathological variables in this patient cohort.

### G-MDSCs from MBC patients have a transcriptome similar to that of mature neutrophils

To investigate the potential origin and identity of human G-MDSCs, we next analyzed the transcriptomes of these and other cells. We profiled gene expression of G-MDSCs sorted from MBC and Gram-positive sepsis patients (from the PBMC fraction), and CD14+ monocytes (HC Mo sample) and CD15+ neutrophils (HC N sample; derived from the whole blood) from healthy donors (sorted as shown in Fig 1C). Hierarchical clustering analysis revealed the presence of three different gene profile clusters, with the expression profile of G-MDSCs from MBC patients most similar to that of neutrophils from healthy donors (HC N) (Fig 2A). As expected, the expression profile of monocytes from healthy donors (HC Mo)

diverged from those of the other analyzed cell populations (Fig 2A). PCA analysis of gene expression clusters confirmed the conclusions of hierarchical clustering analysis (Fig 2B). Examples of differentially expressed genes in the various cell populations are shown in Fig 2C. Surprisingly, the gene expression profile of G-MDSCs from Gram-positive sepsis patients was divergent from that of G-MDSCs derived from MBC patients, although typical G-MDSC–specific genes were expressed in both MBC and sepsis G-MDSCs (Fig 2C). Genes that were expressed at high levels in MBC and sepsis patients compared with HC N cells were relevant to immunosuppression and metastasis (ARG1, MMP8, TLR5, and CD177) (Fig 2C, green). ARG1 and MMP8 are characteristic for G-MDSCs (Gabrilovich, 2017).

Identification of differentially expressed genes between MBC and Gram-positive sepsis G-MDSCs resulted in 1,167 significantly affected genes ($P < 0.05$; ±0.5 $\log_2$-fold-change; Table S1 and Fig 2D). Some selected genes were significantly higher expressed in sepsis as compared with MBC samples (CD24, CD177, MMP8, and TLR5; Table S1 and Fig 2D), and in MBC compared with sepsis (HLA-DRA, CSF1R, and AGAP6/9; Table S1 and Fig 2D). Lox-1 (OLR1), a typical G-MDSC gene, was highly expressed but varied considerably within MBC and sepsis samples (Table S1). Of note, various genes from the same functional categories relevant to tumor growth, such as angiogenesis, metastasis, and immunosuppression, were differentially expressed in the cell populations, suggesting pathway-specific transcriptional programs operational in these cells. Collectively, the above findings indicate that MBC G-MDSCs are more similar to healthy donor neutrophils than to Gram-positive sepsis G-MDSCs, and that both MBC G-MDSCs and Gram-positive sepsis G-MDSCs express typical G-MDSC genes, but also unique genes of potential importance.

### G-MDSCs are neutrophils at distinct maturation stages

An obvious limitation of gene expression profiling of sorted bulk G-MDSCs is that the dominant population in a heterogeneous sample will mask any minor subpopulations present. Therefore, to visualize minor and potentially unique cell subpopulations within G-MDSCs, we next performed mass cytometry (CyTOF) of PBMCs, gating on CD15+ cells (LDGs), comparing healthy controls, patients with MBC, patients with Gram-positive sepsis, and patients with malignant melanoma, as an example of another cancer type (Fig 3A and B). By generating a t-distributed stochastic neighbor embedding (tSNE) plot to visualize differences between cells, we showed that healthy control LDGs were distinct from LDGs from both cancer patient populations, as well as from Gram-positive sepsis LDGs (Fig 3A). Interestingly, we detected distinct subpopulations that were unique to cancer patient LDGs (MBC and malignant melanoma; Fig 3A, black arrows) for each parameter analyzed (Figs 3B and S2). The analysis revealed that markers related to neutrophil maturity (CD10, CD13, and CD45) were down-regulated in these unique subpopulations, indicating the presence of immature neutrophils

staining and IHC. The cells were a morphologically heterogeneous population with blast-like (orange arrow) and PMN (black arrow) nuclei, and, occasionally, banded neutrophils (purple arrow). The frequencies of blasts and PMNs, determined based on the number of cells with the indicated morphology in a microscopy field under 20× magnification is shown. Error bars indicate SEM. *$P < 0.05$; $N = 4$; Mann–Whitney test. **(E)** The sorted G-MDSCs were negative for α-smooth muscle actin and CD68 expression, positive (brown) for CD31 expression, and only sporadically (<1% cells) expressed CD34, as determined by IHC.

**Table 1.  Clinical correlations for patients with "high" or "low" systemic G-MDSCs[a]–G-MDSCs levels do not correlate with disease severity.**

| G-MDSC | "High" | "Low" | P-value |
|---|---|---|---|
| | N = 7 | N = 18 | |
| **Age (y)** | | | |
| <65 | 2 | 8 | 0.66[b] |
| >65 | 5 | 10 | |
| **Performance status (ECOG)** | | | |
| 0 | 3 | 12 | |
| 1 | 1 | 2 | 0.56[c] |
| 2 | 3 | 4 | |
| **Tumor type** | | | |
| Ductal | 5 | 11 | |
| Lobular | 2 | 4 | 0.67[c] |
| Other | 0 | 3 | |
| **NHG** | | | |
| 1 | 0 | 2 | |
| 2 | 4 | 10 | |
| 3 | 1 | 3 | 1.00[c] |
| Unknown | 2 | 3 | |
| **Tumor size (T)** | | | |
| T1 | 5 | 7 | |
| T2 | 1 | 5 | |
| T3 | 0 | 3 | 0.65[c] |
| T4 | 1 | 3 | |
| **Node status (N)** | | | |
| N+ | 3 | 13 | |
| N– | 4 | 4 | 0.17[c] |
| Unknown | 0 | 1 | |
| **Adjuvant chemotherapy** | | | |
| Yes | 2 | 9 | |
| No | 5 | 9 | 0.41[c] |
| **Adjuvant endocrine therapy** | | | |
| Yes | 3 | 15 | |
| No | 4 | 3 | 0.07[c] |
| **Breast cancer Subtype[d]** | | | |
| ER$^+$HER2$^-$ | 3 | 15 | |
| HER2$^+$(ER$^{+/-}$) | 1 | 1 | |
| TNBC (ER$^-$HER2$^-$) | 2 | 2 | 0.16[c] |
| Unknown | 1 | 0 | |
| **Metastasis-free interval (y)** | | | |
| 0 | 1 | 1 | |
| >0–3 | 0 | 3 | 0.57[c] |
| >3 | 6 | 14 | |
| **Number of metastatic sites** | | | |

**Table 1. Continued**

| G-MDSC | "High" | "Low" | P-value |
|---|---|---|---|
| | N = 7 | N = 18 | |
| 0–2 | 6 | 12 | 0.63[c] |
| 3–5 | 1 | 6 | |
| Metastatic site | | | |
| Lymph nodes versus not | 2/5 | 7/11 | 1.0[c] |
| Lung versus not | 2/5 | 9/9 | 0.41[c] |
| Liver versus not | 1/6 | 4/14 | 1.0[c] |
| Bone versus not | 5/2 | 14/4 | 1.0[c] |
| Visceral versus not | 4/3 | 12/6 | 0.67[c] |
| Bone-only versus not | 1/6 | 4/14 | 1.0[c] |
| Number of CTCs | | | |
| ≥5 | 3 | 9 | 1.0[c] |
| <5 | 4 | 9 | |
| Peripheral neutrophil count | | | |
| Median (range) | 7.20 (4.00–12.20) | 4.00 (2.00–10.20) | <0.05[e,f] |

[a]The highest level of healthy control G-MDSCs were set as normal range value (10.5% of PBMCs). "Low" G-MDSCs were patients with G-MDSCs below and up to normal range (≤10.5%). "High" G-MDSCs were patients with G-MDSCs levels above the healthy control normal range (>10.5%).
[b]P-value from Pearson's chi-squared test.
[c]P-value from Fisher's Exact Test.
[d]Breast cancer subtype was primarily derived from immunohistochemical staining of the metastasis. If no information was available from the metastasis, the subtype was derived by staining of the primary tumor.
[e]P-value from Mann Whitney test.
[f]P < 0.05.

(Bergenfelz & Leandersson, 2020). Collectively, the analysis indicated that a subgroup of G-MDSCs have a distinct profile in cancer patients compared with cells from sepsis patients and healthy controls. However, given the limited number of patients in this analysis, the level of inter-patient heterogeneity, rather than disease-specific context, is difficult to truly establish at this stage. The peripheral LDG population in cancer patients did, however, comprise neutrophils at different maturation stages, with the majority being mature activated neutrophils (because of their low density) and the minority being immature neutrophil subpopulations at varying maturation stages. These data support the emergency myelopoiesis model of G-MDSC generation (Pillay et al, 2013).

### G-MDSCs from MBC patients affect tumor growth in vivo

The function of G-MDSCs has historically been evaluated using mouse G-MDSCs or in vitro models. To investigate a potential role of human G-MDSCs in tumor progression in vivo, we co-transplanted primary human G-MDSCs ($2 \times 10^5$ cells/animal) from MBC patients with the human BC cells MDA-MB-231, subcutaneously, in severely immuno-deficient Nod scid gamma (NSG) mice (Shultz et al, 2005). Control mice received BC xenograft only. NSG mice lack functional lymphocytes, with defective macrophages and dendritic cells, as a consequence of common γ chain ($\gamma_c$) deletion, but produce monocytes and some neutrophils, and allow multi-lineage human hematopoietic stem cell engraftment (Shultz et al, 2005). Transplanted tumors were allowed to develop for 21 d and tumor biological characteristics were analyzed,

including growth, cell proliferation, angiogenesis, stroma formation, and the presence of myeloid cells of mouse and human origin (Figs 4 and S3A–C). Xenografts of primary human G-MDSCs co-transplanted with BC cells (G-MDSC/BC), were significantly larger (Fig 4A), accompanied by a slight increase in the expression of proliferation marker Ki67 (Fig 4B), than xenografts containing only BC cells (Fig 4A and B). That G-MDSCs affect tumor cell proliferation was in line with what has previously been reported for mouse G-MDSCs (Yang et al, 2004; Ouzounova et al, 2017). The proliferation of MDA-MB-231 cells in in vitro co-culture with human neutrophils (LDGs, representative of activated neutrophils and high-density granulocytes [HDGs], representative of mature neutrophils, from HC) was, however, not affected (Fig S4A). This indicated that either the effect of G-MDSCs on tumor cell proliferation is dependent on other mediators in vivo, or that patient-derived G-MDSCs have unique effector mechanisms as compared with LDGs.

To evaluate the survival of human G-MDSCs within the G-MDSC/BC xenografts, we analyzed the expression of the human-specific pan-myeloid cell marker CD11b, anti-inflammatory myeloid cell marker CD163, and MDSC myeloid cell marker S100A9 (Allaoui et al, 2016) in xenografts. These human myeloid markers were not detectable by IHC (Fig S3B and C). This indicates that the transplanted G-MDSCs, independent of their maturity, did not survive 21 d of transplantation. This outcome clearly contrasts to transplanted monocytes that we have previously shown survive transplantation up to 90 d (Allaoui et al, 2016). Similarly, G-MDSCs did not significantly affect tumor stroma formation, as determined by histological

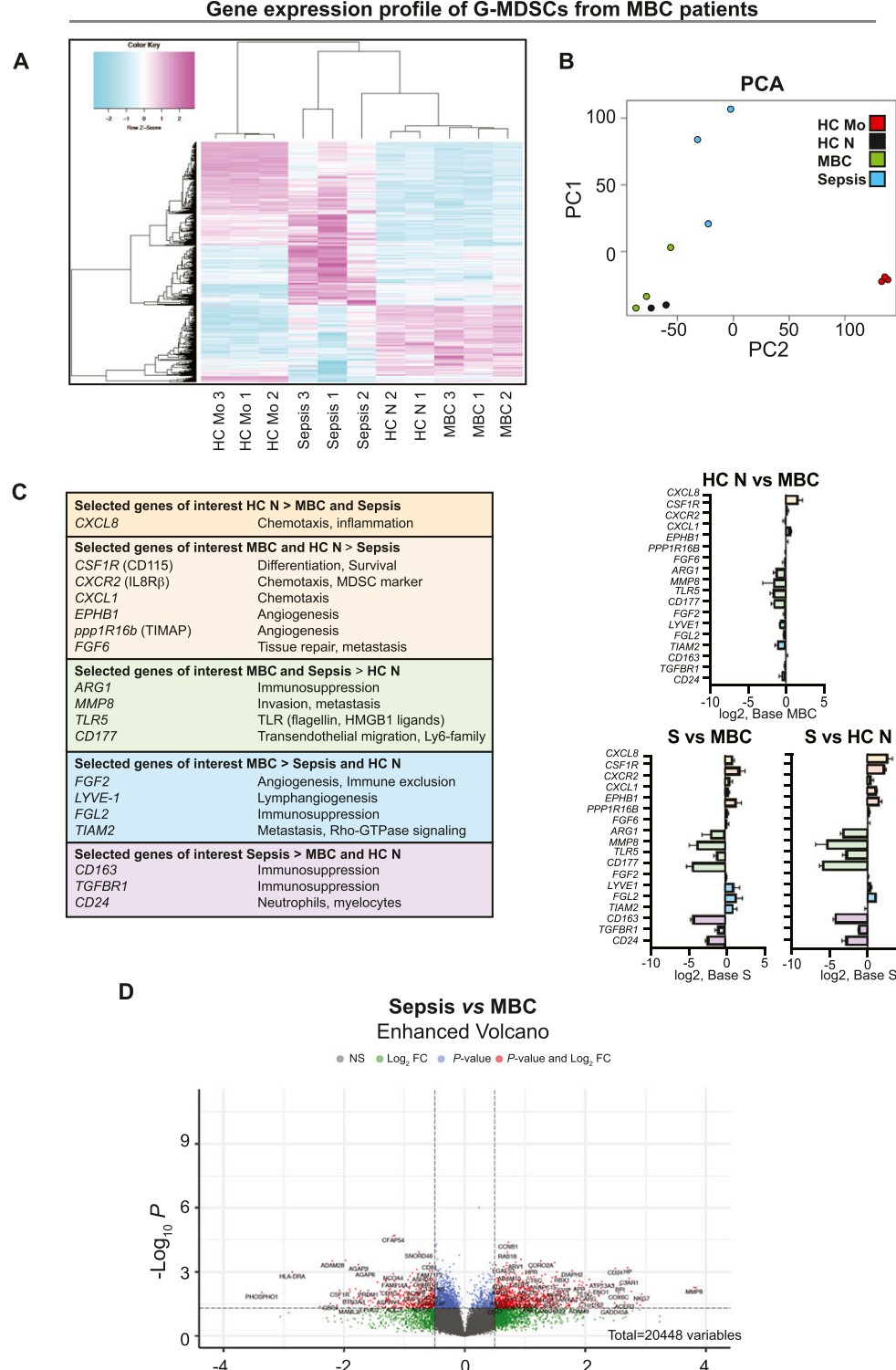

Figure 2. Gene expression profiles of granulocytic myeloid-derived suppressor cells (G-MDSCs) from metastatic breast cancer (MBC) patients are similar to those of neutrophils from healthy donors.
**(A)** Hierarchical clustering (method average) of gene expression profiles of MBC patient G-MDSCs (MBC 1–3), sepsis patient G-MDSCs (Sepsis 1–3), healthy control neutrophils (HC N 1–2; sample number 3 was excluded because of low RNA yield), and healthy donor monocytes (HC Mo 1–3). Probe sets with a fold-change expression >2 and $P < 0.05$ between the Sepsis (S) and MBC groups were plotted in Heatmap2 in R. **(B)** PCA diagram of overlap and significant differences between gene profiles, showing high similarity between MBC patient G-MDSCs (green) and healthy control neutrophils (black), low similarity with Gram-positive sepsis patient G-MDSCs (blue), and no similarity with HC Mo (red). **(C)** Expression patterns of selected genes. Because $N = 2$ for HC N samples, no statistical analysis of data was performed in relation to HC N. **(D)** Volcano plot showing significant differences between sepsis (S) and MBC gene profiles.

staining with Sirius Red to detect the classical collagen types I, III, and IV and by using the α-SMA stain to determine the amount of activated myofibroblasts (Figs 4C and S3A). This was also in contrast with the effect of transplanted monocytes (Allaoui et al, 2016).

In summary, the above observations indicate that transplanted patient–derived G-MDSCs affect tumor growth by promoting cell proliferation in vivo and that this, due to low survival of G-MDSCs, likely occurs early in tumor progression.

**A**

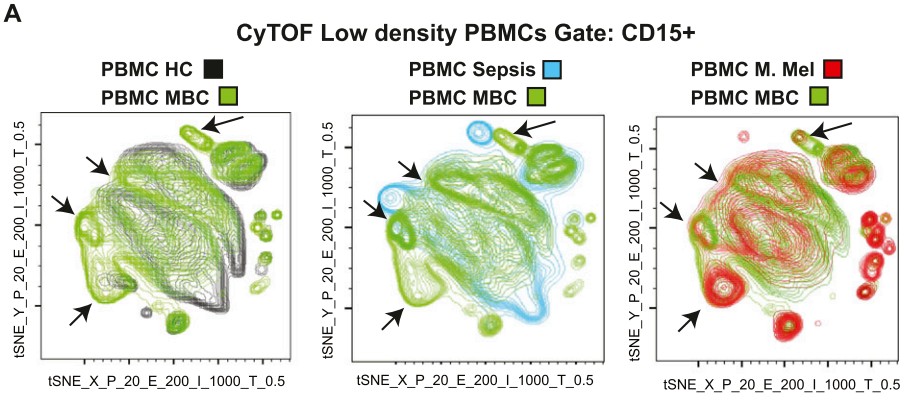

## CyTOF Low density PBMCs Gate: CD15+

**B**

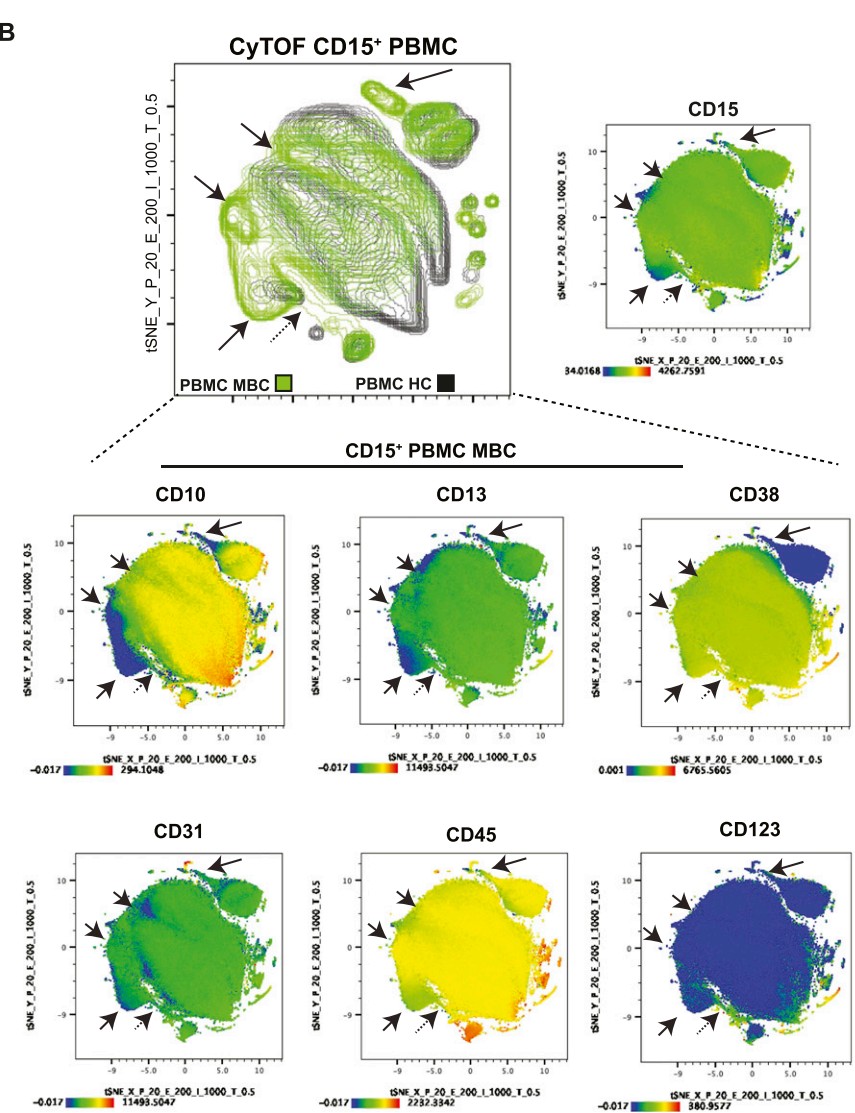

**Figure 3. CyTOF analysis of granulocytic myeloid-derived suppressor cells from metastatic breast cancer (MBC) patients reveals unique cell populations.**

**(A)** Mass cytometry analysis of PBMCs from healthy donors (black *N* = 2), and patients with MBC (green *N* = 2), malignant melanoma (M. Mel, red *N* = 2), and Gram-positive sepsis (blue *N* = 1). The tSNE plots show CD15$^+$-gated cells (low density granulocytes). The black arrows indicate unique populations associated with cancer patients (MBC and malignant melanoma) that do not overlap with healthy control or sepsis patient CD15$^+$ PBMCs. **(B)** PBMCs from healthy donors (black) and MBC (green) patients were analyzed by CyTOF. The tSNE plot represents CD15$^+$-gated cells. The black arrows indicate unique populations associated with cancer patients (MBC and malignant melanoma) that do not overlap with healthy control or sepsis patient CD15$^+$ PBMCs. Each plot below the tSNE plot represents a marker, as indicated (see Fig S2 for additional markers); blue color indicates low expression and red color indicates high expression of the indicated marker, in all CD15$^+$-gated cells. Data analysis: 39-parameter data experiments in FlowJo version 10; Perplexity = 20, Eta = 200, Iter = 1,000, Theta = 0.5. Output event = 424,898, down sampled to 30,000 cells.

## G-MDSCs from MBC patients affect vessel formation in vivo

To further investigate why the G-MDSC co-transplanted tumors were larger than BC tumors, we next analyzed the formation of blood and lymph vessels in the xenografts. Specifically, we stained the tumor tissue for the presence of endothelial cell marker CD31 of both human and mouse origin, and mouse Lyve-1, a protein found on lymphatic endothelial cells (Kong et al, 2017). We did not detect

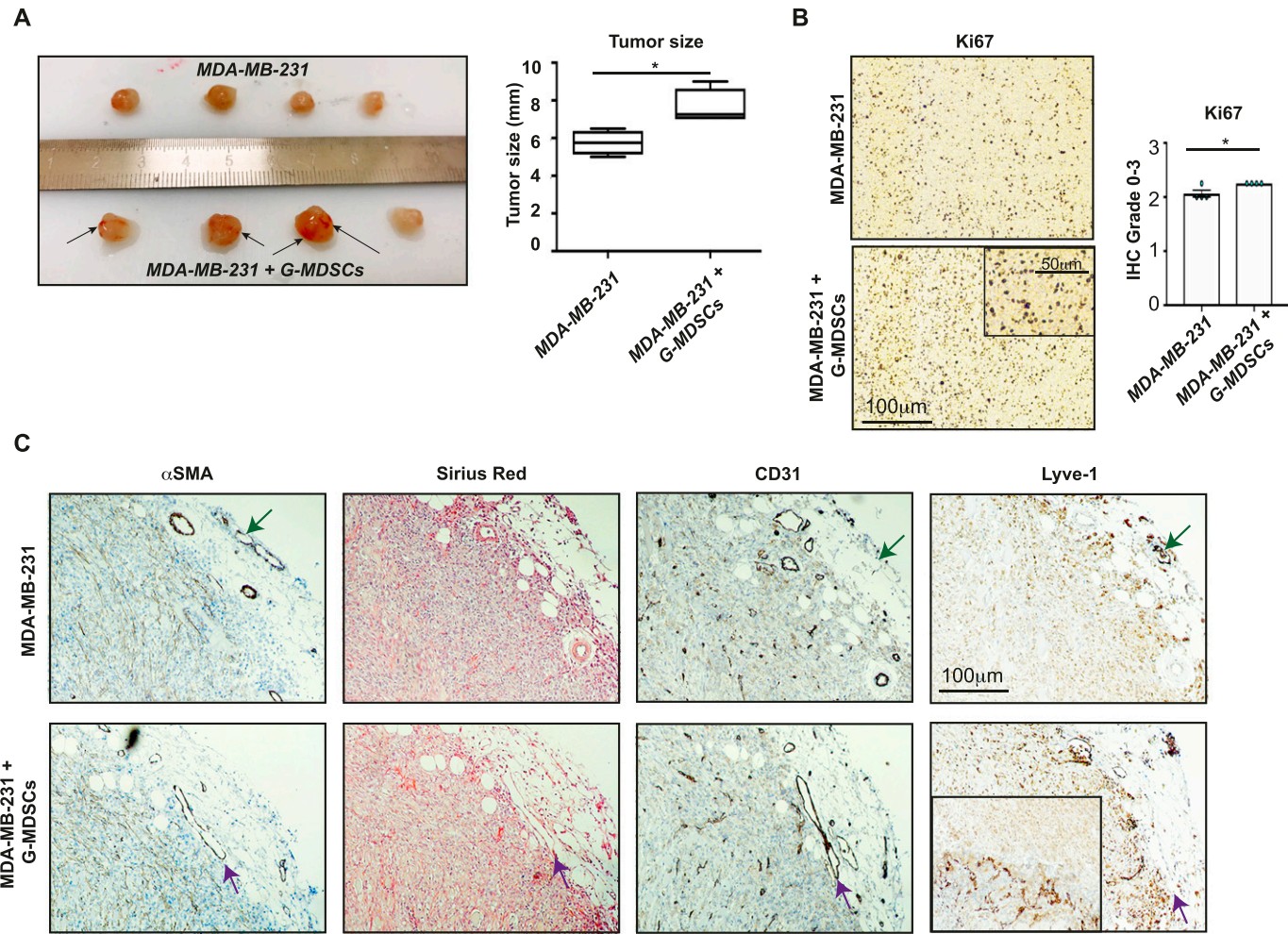

**Figure 4. Granulocytic myeloid-derived suppressor cells (G-MDSCs) from metastatic breast cancer patients co-transplanted with MDA-MB-231 breast cancer cells promote tumor growth and affect vessel formation in vivo.**
G-MDSCs from metastatic breast cancer patients (2 × 10⁵ cells/mouse) were co-transplanted with MDA-MB-231 breast cancer cells (2 × 10⁶ cells/mouse) (BC) in immunodeficient Nod scid gamma mice for 21 d. **(A)** Xenograft tumors consisting of G-MDSC/BC cells were significantly larger than xenograft tumors consisting of only BC cells. $N = 4$. The black arrows indicate possible sites of hemorrhages, indicating leakage of blood vessels in the G-MDSC co-transplanted tumors only. Error bars indicate SEM. *$P < 0.05$; Mann–Whitney U-test. **(B, C)** Tumor sections were stained for the presence of the indicated markers: proliferation marker Ki67, endothelial cell marker CD31, lymphatic endothelial cell marker Lyve-1, activated fibroblast marker α-smooth muscle actin, and the collagen marker Sirius Red. $N = 4$. Error bars indicate SEM. *$P < 0.05$; Unpaired $t$ test. **(C)** Statistical evaluation of expression levels and vessel area from (C) is shown in Fig S3A.

human CD31 in any xenografts, ruling out trans-differentiation of myeloid cells into endothelial cells or contamination of the G-MDSC sample with human bone marrow–derived mesenchymal stem cells (Fig S3B and C). Mouse CD31 was expressed on endothelial cells in both BC and G-MDSC/BC xenografts. Furthermore, the overall number of blood vessels did not differ significantly between the xenografts (Figs 4C and S3A). However, the G-MDSC/BC xenografts contained slightly larger vessels (>70 $\mu m^2$) with a defined lumen than the blood vessels present in the control BC xenografts (Figs 4C and S3A), possibly with a slightly lower level of PDGFRβ expression in the G-MDSC transplanted xenograft pericytes, although not significant (Fig S3B). In parallel, the amount of Lyve-1⁺ lymph vessels was marginally increased in G-MDSC/BC xenografts (Figs 4C and S3A). Finally, the arrows in Fig 4A indicate possible sites of hemorrhages, indicating leakage of blood vessels in the G-MDSC co-transplanted tumors only. Hence, we inferred that human G-MDSCs

could affect the size of vessels formed in tumors and, possibly, also the lymphangiogenic potential.

Accordingly, we next examined the angiogenesis-regulatory potential of MBC G-MDSCs, using gene set enrichment analysis (GSEA) of angiogenesis-related genes from the gene expression profiling. We identified unique angiogenesis-related genes in MBC G-MDSCs that potentially could be involved in the regulation of vessel lumen formation (Fig S4B and C and Table S1). Among the genes that were expressed at a significantly higher level only in MBC G-MDSCs compared with HC-Mo were those encoding angiogenesis inhibitors *SPINK5*, *COL4A2*, *COL4A3*, and *NOTCH4*; the angiogenesis regulator *AMOT*; and the proangiogenic factors *PROK2* and *SHH*. Collagen IV is visualized by Sirius Red staining (detecting collagen I, III, and IV) in the xenografts (Fig 4C). Together, this indicates that G-MDSCs from MBC patients may be involved in the regulation, or even dysregulation, of angiogenesis, potentially explaining the

slightly increased lumen size of the vessels in G-MDSC/BC xeno-grafts (Fig S4B and C and Table S1).

## G-MDSCs from MBC patients promote myeloid immune cell exclusion in vivo

To investigate whether the G-MDSC/BC tumor size was increased because of enhanced infiltration of mouse myeloid cells per se, we next stained the tumor tissue for the presence of mouse monocytes (Ly6C$^+$ cells) or neutrophils (Ly6G$^+$ cells) (Figs 5A and S3A and C). All xenografts were Ly6G negative (Fig S3B and C). Only small amounts of mouse myeloid cells of the monocytic/macrophage lineage (Ly6C$^+$ or F4/80$^+$) were present in the BC xenografts, and these cells were not detected in the G-MDSC/BC xenografts (Figs 5A–C, S3B and C, and S5A and B). Surprisingly, as determined by IHC, the Ly6C expression was primarily localized to endothelial-like structures (Jutila et al, 1988) in the BC xenografts (Figs 5A, S3B and C, and S5A). The role of endothelial cell Ly6C is not well examined, but it is known that Ly6C is crucial for the trans-endothelial migration of resident lymph-node memory CD8$^+$ T cells (Jaakkola et al, 2003; Hanninen et al, 2011). Using immunofluorescence (IF), which is more sensitive than IHC, we were able to detect some Ly6C expression in the G-MDSC/BC xenografts (Fig 5). However, Ly6C/CD31 double staining revealed that Ly6C was significantly more often co-expressed with CD31 (representing the endothelial cells) in BC xenografts than in G-MDSC/BC xenografts (Figs 5B and C, S3B and C, and S5A and B). It was also evident that myeloid cells of the monocytic lineage (Ly6C$^+$) did not extensively infiltrate the tumor in G-MDSC/BC xenografts, even though they were visualized inside the vessel lumen in these xenografts (Figs 5B and C, S3B and C, and S5A and B). This suggests that human G-MDSCs affect both expression of trans-endothelial adhesion molecules (Ly6C) and myeloid cell tumor infiltration.

To investigate whether the Ly6C levels in endothelial cells were affected by G-MDSC–like cells in vitro, we co-cultured mouse en-dothelial MS1 cells with primary human LDGs and HDGs from healthy donors. Only primary human LDGs decreased the expression of *Ly6C* mRNA significantly in mouse MS1 endothelial cells (Fig 5D), in an ROS-independent manner (Fig S6A), indicating that activated neutrophils also have this potential. Together, this further confirm our in vivo findings that G-MDSCs are directly associated with re-duced expression of Ly6C on endothelial cells and reduced infil-tration of myeloid immune cells.

## G-MDSC–like cells (LDGs) reduce endothelial expression of CX3CL1

In an immunologically hot tumor, immune cells are attracted to the inflamed tumor vessels by endothelial-derived chemokines before trans-endothelial migration (Lee et al, 2018). The function of chemokines could be negatively regulated via tyrosine nitration (nitrotyrosine formation) by ROS (Thompson et al, 2017). We therefore investigated nitrotyrosine levels in the G-MDSC/BC xe-nografts to determine if reduced myeloid immune cell infiltration is caused by ROS-induced nitrotyrosine. Compared with BC xeno-grafts, nitrotyrosine levels were not increased in the G-MDSC/BC xenografts (Fig S3C). Recently, CX3CL1 (fractalkine) was recognized as the principal endothelial-derived chemokine responsible for

turning immunologically cold tumor into a hot tumor (Lee et al, 2018). In immunologically cold tumors, angiogenic growth factors, such as bFGF and VEGF, down-regulate CX3CL1 (Sidibe et al, 2018). We therefore tested whether human G-MDSCs impacted CX3CL1 expression in mouse endothelial cells. Indeed, upon co-culture of MS1 cells with primary human G-MDSC–like cells (LDGs), but also with HDGs, the *Cx3cl1* mRNA levels in mouse MS1 endothelial cells decreased (Fig 5E). In line with this, we detected a modest inverse correlation between the expression of *CX3CL1* and that of the typical G-MDSC gene *OLR1* (Condamine et al, 2016) in primary human BCs (Fig S6B). Collectively, the findings described above suggest that both human G-MDSCs and neutrophils reduce the expression of Ly6C and CX3CL1 in endothelial cells, thus potentially inhibiting the infiltration of mouse myeloid immune cells into xenografts.

## G-MDSCs in human breast tumor tissue as defined by CD15$^+$ blasts

Reliable clinical prognostic data are lacking to determine whether the infiltrating human G-MDSCs affect tumor progression or act only on systemic peripheral blood cells, as no bona fide IHC G-MDSCs markers currently exist (Elliott et al, 2017). Furthermore, TANs (CD15$^+$ cells) were shown to infiltrate human BC, but with contradicting prognostic results (Sozzani et al, 2008; Koh et al, 2013). We thus investigated the prognostic weight of CD15$^+$ cells with unique morphologies (immature blast–like versus PMN) in more detail. Using morphology (immature blast–like versus PMN) in combina-tion with the neutrophil activation state (myeloperoxidase [MPO]$^+$ for activated neutrophils), we analyzed a primary BC patient cohort ($n$ = 144). Using double IHC staining, we annotated CD15$^+$ cells with mature neutrophil morphology (CD15$^+$MPO$^+$, i.e., PMN and co-expressing myeloperoxidase MPO) or CD15$^+$ cells with immature myelocyte morphology and no MPO expression (CD15$^+$MPO$^-$, i.e., blast and MPO$^-$ cells) (Fig 6A and B). The majority of CD15$^+$ cells were mature neutrophils (Fig 6A, black arrows); 67% of tumors were infiltrated by CD15$^+$ cells (96 of 144 tumors; Fig 6B, black arrows). Furthermore, 53% of tumors were infiltrated by CD15$^+$MPO$^+$ cells (76 of 144 tumors). Immature neutrophils with a blast-like morphology (CD15$^+$ MPO$^-$) constituted a minority of CD15$^+$ cells (Fig 6B, yellow arrows); 14% of tumors had CD15$^+$ MPO$^-$ blasts (21 of 144 tumors).

The overall infiltration of CD15$^+$ cells (0–3) was associated with ERα negativity ($P$ = 0.048), triple-negative breast cancer (TNBC) subtype ($P$ = 0.018), and also the presence of the anti-inflammatory myeloid cell marker CD163 ($P$ = 0.020), and the MDSC and neutrophil marker S100A9 ($P$ = 0.019) (Table 2). It was however not associated with the pan-macrophage marker CD68 ($P$ = 0.150) (Table 2). After subdividing the infiltrating neutrophils into subsets, we observed that infiltration of CD15$^+$MPO$^-$ blasts (immature neutrophils) was associated with the TNBC subtype ($P$ = 0.037), whereas that of CD15$^+$MPO$^+$ cells (mature activated neutrophils) was not (Table 2). CD15$^+$MPO$^-$ blasts were also associated with the MDSC neutrophil marker S100A9 ($P$ = 0.028), but not with any other clinical or immune cell marker investigated (Table 2). Infiltration of CD15$^+$MPO$^+$ cells was not associated with any marker, although it showed an as-sociation trend with CD163 ($P$ = 0.089). In line with this, patients with tumors infiltrated by CD15$^+$MPO$^-$ blasts showed a trend toward shorter recurrence-free survival than those with tumors infiltrated by all CD15$^+$ or CD15$^+$MPO$^+$ cells (Fig 6C–E). Nevertheless, none of the

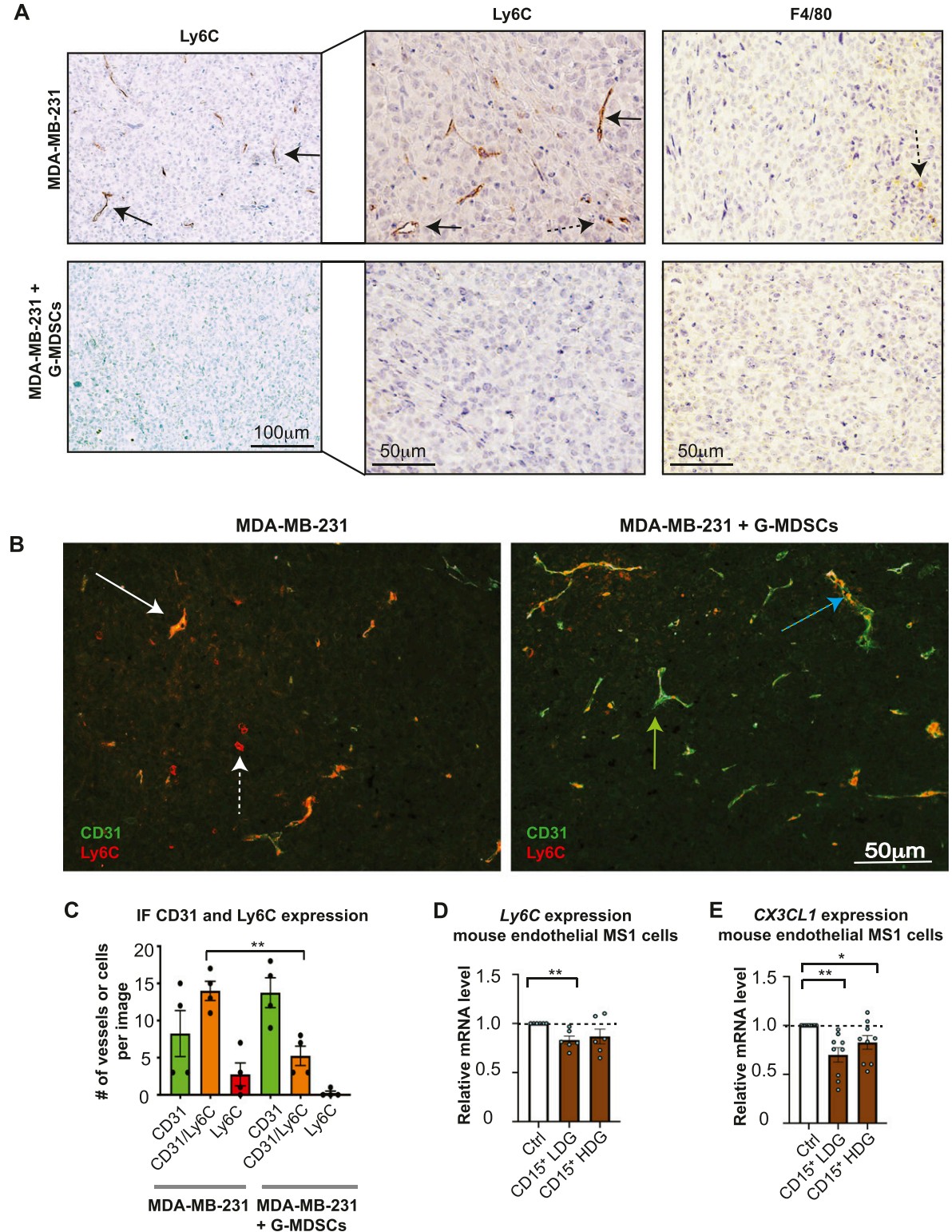

**Figure 5. Human granulocytic myeloid–derived suppressor cells (G-MDSCs) reduce endothelial Ly6C expression and promote immune cell exclusion in mouse model.**
**(A)** IHC analysis of Ly6C levels in xenografts from MDA-MB-231 cells (top) and in MDA-MB-231 grafts co-transplanted with human G-MDSCs (bottom) in Nod scid gamma mice for 21 d. Statistical evaluation of data is shown in Fig S3A. F4/80 mouse macrophage staining is shown as control. **(B, C)** Ly6C and CD31 IF analysis in xenografts from MDA-MB-231 cells (left) and MDA-MB-231 grafts co-transplanted with human G-MDSCs (right). **(C)** Green arrow, endothelial vessel (CD31) with no Ly6C co-staining; white solid arrow, endothelial vessel (CD31) co-stained for Ly6C; white dashed arrow, Ly6C$^+$ CD31$^-$ myeloid cells infiltrating the tumor; blue dashed arrow, endothelial vessel (CD31) with potential Ly6C$^+$ cells in the vessel lumen (red inside green in C). Statistical evaluation (C) of the IF data from the different xenografts is shown (green = only CD31$^+$ vessels; orange = CD31$^+$Ly6C$^+$ vessels; red = infiltrating myeloid Ly6C$^+$ cell). N = 4. Error bars indicate SEM. **P < 0.01; unpaired t test. **(D, E)** Relative mRNA levels of mouse Ly6C and CX3CL1 in mouse endothelial MS1 cells co-cultured for 48 h with human CD15$^+$ low-density granulocytes, and in untreated MS1 control cells. N = 5. *P < 0.05, **P < 0.01; paired t test.

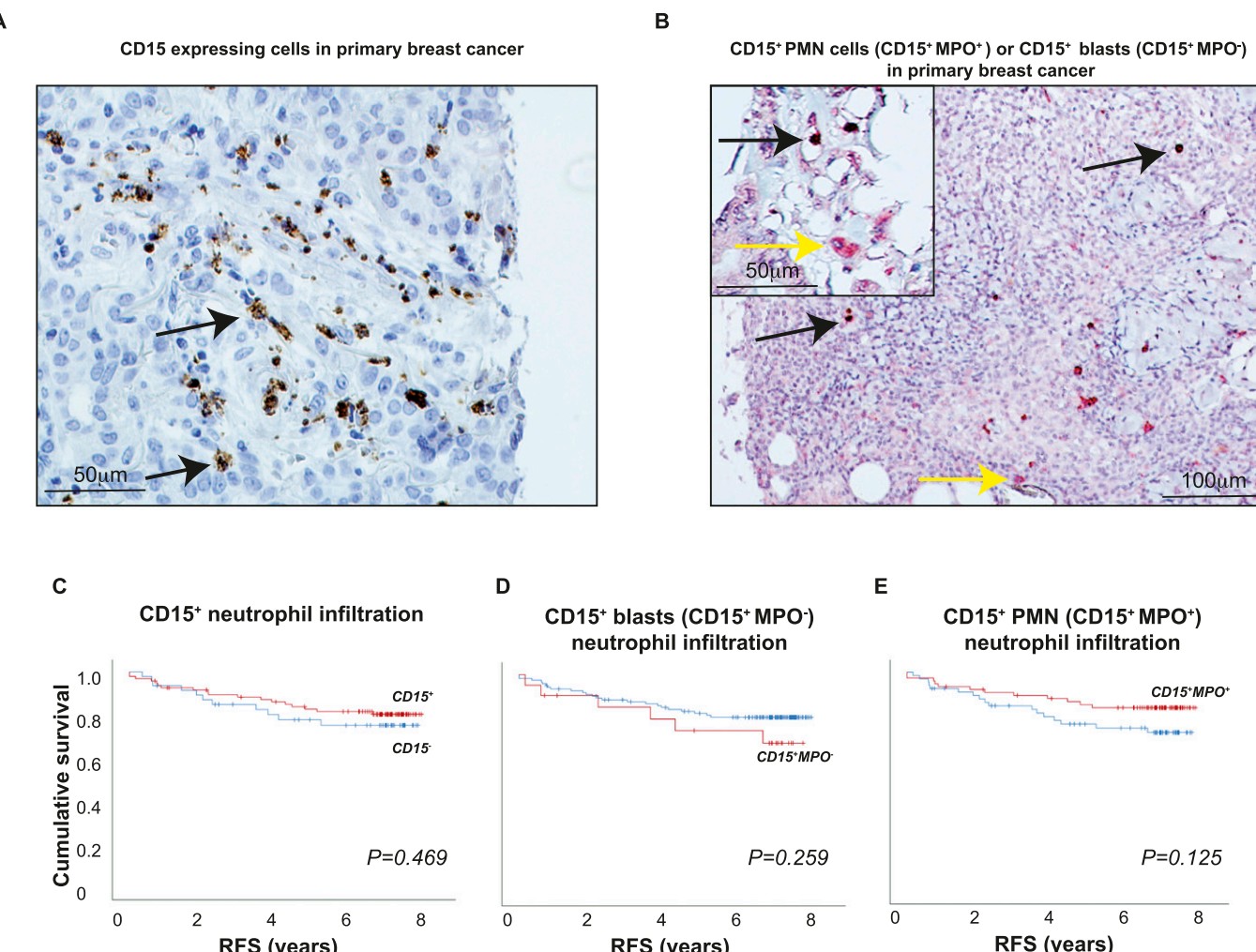

**Figure 6.   IHC evaluation of CD15⁺, CD15⁺MPO⁺, and CD15⁺MPO⁻ cells in a clinical cohort of primary breast cancers.**
**(A)** CD15⁺ cells are present in human primary breast tumors (black arrows). **(B)** CD15⁺PMN⁺MPO⁺ (CD15⁺MPO⁺) representing mature neutrophils or PMNs (black arrows), and CD15⁺PMN⁻MPO⁻ (CD15⁺MPO⁻) cells with a blast-like morphology (yellow arrows) are found in primary human breast tumors. Representative pictures are shown. **(C, D, E)** Recurrence-free survival of patients, and infiltration of CD15⁺ cells with or without the mature neutrophil marker MPO and PMN morphology in tumors. A clinical cohort of 144 breast cancer patients was evaluated. CD15⁺MPO⁺ cells were considered to be mature neutrophils; CD15⁺MPO⁻ cells with blast-like immature morphology were considered to be immature neutrophils. Log-rank $P < 0.05$ was considered significant.

infiltrating CD15⁺ cells significantly affected the survival or BC recurrence (Fig 6C–E). In summary, it is conceivable that whereas the mature neutrophils represent the most typical neutrophil variant in BC, the CD15⁺MPO⁻ immature neutrophils are associated with S100A9 and worse prognosis, although this correlation was not significant. In that respect, they are more similar to G-MDSCs than the mature CD15⁺MPO⁺ infiltrating cells.

# Discussion

Although the immunosuppressive role of MDSCs is well established, the identity and specific activities of human G-MDSCs are elusive and have not been determined in vivo. We show that human G-MDSCs represent neutrophils at distinct maturation stages, with

the surface profile specific to the host's malignancy. Furthermore, we demonstrate that human G-MDSCs from MBC patients promote tumor growth and affect vessel formation in vivo, in addition to exerting immune cell excluding effects.

We found that the gene expression profile of G-MDSCs from MBC patients was most similar to that of whole blood neutrophils from healthy donors, representing neutrophils of both resting (HDGs) and activated (LDGs) nature. Surprisingly, the gene expression profile of G-MDSCs from MBC patients differed from that of cells derived from Gram-positive sepsis patients. This was supported by the results of mass cytometry analysis. In contrast to the prolonged chronic inflammation observed in cancer patients, sepsis is characterized by acute inflammation. This probably leads to completely different pathways generating and activating G-MDSCs, involving pattern-recognition receptor activation by pathogen-associated molecular patterns in sepsis and damage-associated

**Table 2. Association between the presence of CD15 (0–3), CD15⁺MPO⁺ (0, 1), or CD15⁺MPO⁻ (0, 1) tumor-infiltrating cells, and clinical and immunological parameters in primary breast cancer.**

| Clinical and immunopathological features | | CD15 (0–3) | | CD15⁺ (PMN/MPO)⁺ | | CD15⁺ (PMN/MPO)⁻ | |
|---|---|---|---|---|---|---|---|
| Pearson's chi-squared test[a,b] | (N) | Correlation Coefficient[a] | P-value | Correlation Coefficient[a] | P-value | Correlation Coefficient[a] | P-value |
| ERα[a] | (143) | −0.165 | **0.048**[c] | 0.004 | 0.962 | −0.130 | 0.122 |
| PR[a] | (143) | −0.004 | 0.958 | 0.133 | 0.113 | −0.110 | 0.188 |
| Her2[a] | (138) | −0.021 | 0.808 | −0.128 | 0.135 | 0.134 | 0.115 |
| TNBC[a] | (138) | 0.201 | **0.018**[c] | 0.003 | 0.972 | 0.177 | **0.037**[c] |
| CD163 (M2 TAMs, MDSCs (0, 1))[a] | (105) | 0.226 | **0.020**[c] | 0.167 | 0.089 | 0.018 | 0.858 |
| S100A9 (MDSCs (0–4))[a] | (134) | 0.152 | 0.079 | −0.001 | 0.994 | 0.144 | 0.098 |
| S100A9 (MDSCs (0–4))[b] | (134) | | **0.019**[c] | | 0.466 | | **0.028**[c] |
| CD68 (TAMs (0, 1))[a] | (107) | 0.140 | 0.150 | 0.089 | 0.362 | 0.034 | 0.723 |

[a]Bivariate Pearson's chi-squared.
[b]Pearson's chi-squared.
[c]P < 0.05.

molecular patterns in combination with tumor-derived factors, such as colony-stimulating factors, in cancer (Millrud et al, 2017; Bergenfelz & Leandersson, 2020). This might explain the unique immature cancer G-MDSC profile of the released neutrophil pool. The significant differences in gene expression in MBC as compared with sepsis samples, such as *HLA-DR* and ArfGAP proteins (*AGAP*) representing antigen presentation and vesicle transport, indicate unique biological functions that may be of importance for G-MDSCs in cancer (Gamara et al, 2015; Lin & Lore, 2017). A drawback to this study is obviously the limited number of patients involved, thus the level of inter-patient heterogeneity, rather than disease-specific context, is difficult to truly establish at this stage.

Although similarities between G-MDSCs and activated neutrophils have been reported, including gene profiles, morphological, and immunosuppressive features (Rotondo et al, 2009; Pillay et al, 2013), these observations are also in contrast with other studies (Condamine et al, 2016; Marini et al, 2017). In those studies, conventional activated neutrophils (Lox1⁻ LDGs) where found to be immunostimulatory and, therefore, Lox1⁺ G-MDSCs were proposed to be neutrophils of a distinct immunosuppressive activation stage (Condamine et al, 2016), whereas in the other study, immature CD10⁻ neutrophils within the LDG fraction were found to be immunostimulatory, and the mature LDGs were not (Marini et al, 2017). Moreover, in patients with autoimmune diseases, LDGs are immunostimulatory (Silvestre-Roig et al, 2016). The major reasons for all discrepancies could be the methodologies used, but because both immature, activated and also aging neutrophils may be present in the same LDG fraction of cells (Silvestre-Roig et al, 2016; Mackey et al, 2019), the more likely cause is the type of disease, including tumor type, stage, and affected organs. In the present study, gene expression profiling was performed on sorted bulk G-MDSC populations, using different assays (Affymetrix and CyTOF), and importantly using a different control group (Gram-positive sepsis patients). The latter explanation regarding type of disease would concur with our data, where sepsis G-MDSCs differ from MBC G-MDSCs.

In the present study, we show that G-MDSCs from MBC patients comprise a morphologically heterogeneous population containing both blast-like myelocytes and typical PMN cells representing mature neutrophils. The presence of myelocytes among the PMN cells in sorted G-MDSCs has been described before, both in cancer and sepsis patients (Janols et al, 2014; Sagiv et al, 2015; Millrud et al, 2017). Indeed, the clinical term "left shift" has long been used to describe the enrichment of immature neutrophils (left-hand side in Fig 1A) in patients suffering from infectious diseases (Christensen et al, 1981). It is intriguing to speculate whether the blast-like cells have different functions depending on disease (Condamine et al, 2016; Marini et al, 2017) or whether they comprise the bona fide G-MDSC population. Perhaps, the ratio between the blast-like cells and PMNs fluctuates during the course of disease, or with differences in patient microenvironment, reflecting emergency myelopoiesis by various mechanisms. These findings should be confirmed by evaluating single cell multi-omics in larger studies with more patient groups.

We found that the G-MDSCs levels in MBC patients were significantly higher than those in healthy donors. However, this difference was not associated with disease severity or any other clinical parameter, other than high peripheral neutrophil counts. This is in contrast with our previous observations that Mo-MDSCs levels correlate with disease progression and severity in a similar patient cohort (Bergenfelz et al, 2015b, 2020). It also contrasts with what has been reported for bladder cancer, hepatocellular cancer, and malignant melanoma (Zhou et al, 2018), where G-MDSC levels correlated with disease severity. Similarly, in a large meta-analysis of patients with solid tumors, increased levels of both Mo-MDSCs and G-MDSCs in pretreatment peripheral blood were shown to correlate with prognostic features and a worse overall survival (Wang et al, 2018). The findings of the present study could be explained by either the cohort size, the possibility that G-MDSCs exert a pro-tumorigenic effect early in disease (also supported by the NSG model data) and therefore do not correlate with the severity late in the disease (MBC), or that G-MDSC generation in the peripheral blood responds to minute signals that are different from those that Mo-MDSCs respond to, thereby reflecting the metastatic disease overall. Either way it would be in line with the assumption

that G-MDSCs accumulate as a consequence of a gradually, chronically affected myelopoiesis (Gabrilovich, 2017). Considering the relatedness between G-MDSCs and mature activated neutrophils, however, neutrophil count in the peripheral blood of cancer patients also correlates with adverse prognosis (Zhou et al, 2018).

Previous work has raised the important question of whether TANs and G-MDSCs in tumors are the same cells, and if they have a prognostic impact, but the answers have been contradictory (Sozzani et al, 2008; Koh et al, 2013). We show here that mature, activated neutrophils represented the predominant neutrophil type in primary human BCs. The intratumoral CD15$^+$MPO$^-$ blast-like cells observed herein may be either immature neutrophil G-MDSCs or N2 type neutrophils because both should be associated with worse prognosis. Indeed, in the mouse, N2 type neutrophils residing in tumors were suggested to have a blast-like morphology (Fridlender et al, 2009). Although the CD15$^+$ activated neutrophils were more frequent in the primary BC patient cohort investigated, only CD15$^+$MPO$^-$ blasts had a negative impact on prognosis, similar to mouse studies (Andzinski et al, 2016), and were associated with the MDSC marker S100A9. This suggests that the immature neutrophil populations in cancer patients have G-MDSC traits and could be considered as a potential therapeutic target in the future.

The specific roles of G-MDSCs in the tumor environment in vivo are not well understood. Using the NSG xenograft model, we showed here that the co-transplanted primary human G-MDSCs affected tumor size and angiogenesis in BC xenografts in vivo, with possible implications for their immunosuppressive/immuno-excluding effect. The observation of very few infiltrating Ly6C$^+$ myeloid cells and a significantly reduced Ly6C expression on endothelial cells in G-MDSC/BC tumors ruled out the possibility that the observed increased tumor size was caused by immune cell infiltration, suggesting a link between dysregulated angiogenesis and tumor growth. Indeed, angiogenesis is fundamental to metastasis (Bacac & Stamenkovic, 2008), and production of molecules by neutrophils and G-MDSCs that affect angiogenesis has been documented (Grecian et al, 2018). GSEA analysis in the present study revealed that genes involved in the regulation of angiogenesis were enriched in MBC G-MDSCs, suggesting that a dysregulated angiogenesis could be associated with the observed vessel lumen size increase and worse pathology in mice with G-MDSC/BC tumors. Expression of the endothelial immune cell attractant *CX3CL1* was also reduced by G-MDSCs in the present study. G-MDSCs produce angiogenic factors, such as bFGF and VEGF-A, which are closely connected with CX3CL1 down-regulation and subsequent immune cell exclusion (Sidibe et al, 2018) and affect endothelial adhesion molecules (Missiaen et al, 2018). This offers a possible mechanism of G-MDSC–affected myeloid immune exclusion, especially because the immunosuppression in situ was not mediated by G-MDSC–derived ROS, as shown by catalase treatment, and lack of nitrotyrosine in the xenografts. Because the NSG mouse is severely immunocompromised, the observed effects of human G-MDSCs are most likely independent of the immunosuppressive mechanisms of other immune cells. It is possible, however, that they evolve in synergy with the immunosuppressive mechanisms of other cell types.

Although we show the immunosuppressive activity of G-MDSCs in vivo, we are unable to unambiguously determine whether all neutrophils in the tumor microenvironment are immunosuppressive. The data presented here do support the key MDSC definition as

immunosuppressive cells (Bronte et al, 2016), but leave a gap in the definitions of G-MDSCs and neutrophils. These obscurities could be resolved by further analyses using multicolor immunohistochemistry, mass cytometry, or using single cell multi-omics platforms, but also by sorting the immature G-MDSCs and performing further in vivo analyses. This would, however, require large amounts of patient-derived peripheral blood neutrophils. Nevertheless, the tumor-promoting functions of immature neutrophils or G-MDSCs should be considered when treating neutropenic cancer patients with G-CSF, which leads to the generation of even more G-MDSCs (Luyckx et al, 2012; Marini et al, 2017). We propose that neutrophils and G-MDSCs, cells that are probably released as a result of emergency myelopoiesis, are problematic but nonetheless essential immune cells, in cancer patients.

In summary, we have presented evidence that human G-MDSCs are generated in patients with metastatic disease, as cells of the neutrophil lineage at a range of maturation stages. The cancer-induced G-MDSCs differ from G-MDSCs isolated from Gram-positive sepsis patients. We have also shown that human G-MDSCs may contribute to tumor progression, by inducing tumor growth, regulating vessel formation, and promoting myeloid immune cell exclusion in vivo. These findings indicate that G-MDSCs affect the tumor environment and should be considered in patients and for choice of therapies. This new information on the role of human G-MDSC in tumor progression in vivo provides a link to previous knowledge on mouse G-MDSC and has implications for human G-MDSC biology in patients, thus significantly moving the field forward.

## Materials and Methods

### Ethics statement

All patient sample collections were approved by the local Regional Ethical Committee in Lund, Sweden. A written informed consent was obtained from patients with Gram-positive sepsis (Dnr 288/2007 and Dnr 2016/340), MBC (Dnr 2016/806, Dnr 2010/135, and Dnr 2011/748), or malignant melanoma (Dnr 191-2007, Dnr 101-2013), and healthy controls (Dnr 2014/669, Dnr 2017/949). Ethical approval for the use of BC specimens (Dnr 447-07) was obtained from the Regional Ethics Committee in Lund, Sweden, whereby patients were offered the option to opt out. Animal procedures (approval M11-15) were approved by the Regional Ethics Committee for Animal Research in Lund/Malmö, Sweden.

### Patient samples

Peripheral blood was collected from patients with newly diagnosed MBC, before starting first-line systemic therapy (for clinical characterization and flow cytometric analyses [Table 1 and Fig 1B] *N* = 25; for all other analyses *N* = 10), patients with newly diagnosed metastatic malignant melanoma and chemotherapy-naïve at the time point of inclusion (*N* = 2), Gram-positive sepsis patients (*N* = 14), and healthy donors. All blood samples were collected in EDTA-containing tubes, handled identically, and analyzed within 24 h.

 **Life Science Alliance**

Some Gram-positive sepsis and healthy control samples had been collected and analyzed previously for other purposes (Janols et al, 2014). The blood was diluted 1:2 in PBS containing EDTA/sucrose, and overlaid on Ficoll-Paque (GE Healthcare) for gradient centrifugation. The primary BC tissue microarray (TMA) cohort, including tumors from 144 primary BC patients at Skåne University Hospital (Borgquist et al, 2008; Rexhepaj et al, 2010) was analyzed by IHC. ER, PR, and Her2 status was analyzed by IHC, as previously described (Rexhepaj et al, 2010), and used as surrogate marker for the molecular subtype (Onitilo et al, 2009). Immunoprofiling of the BC cohort using IHC specific for myeloid cell subtypes was performed previously (Medrek et al, 2012; Bergenfelz et al, 2015a).

### Flow cytometry and cell sorting

PBMCs were immediately stained with antibodies (listed in Table S2) for further flow cytometry analysis and cell sorting. Cells were analyzed using FACSVerse (BD Biosciences) or sorted using FACSAria (BD Biosciences). 7-Aminoactinomycin-D (7AAD) was used as the dead cell exclusion stain (BD Biosciences). G-MDSCs or neutrophils, and monocytes from the same sample, isolated PBMCs from patients, or total blood (after red cell lysis) for neutrophils from healthy controls, were sorted at a purity of >98% based on $CD33^{+/int}$, $CD14^{-/low}$, and $CD64^{-/low}$ (the sorting strategy is indicated by green boxes in Figs 1C and S1A); monocytes were sorted based on $CD33^{+/high}$, $CD14^{+/high}$, and $CD64^{+/high}$ (Fig 1C; orange boxes). The G-MDSCs had a high SSC/low FSC scatter profile (green dashed line, Fig 1C) and were $CD15^+CD33^+CD11b^+CD11c^+CD66b^+CD14^{-/low}HLA-DR^{-/low-}$ $CD127^{-/int}CD123^-CD34^{-/+}$ $\alpha$-SMA$^-$ColI$^-$ (pink boxes in Fig S1A), as previously described (Janols et al, 2014). For the MS1 endothelial co-culture experiments, CD15$^+$ LDGs or HDGs (mature neutrophils) from healthy donors were isolated from PBMCs by Ficoll gradient centrifugation and magnetic cell sorting (MACS) using CD15-microbead isolation kit (Miltenyi Biotec) according to the manufacturer's protocol.

### Mass cytometry

Mass cytometry experiments were performed at the Science for Life Laboratory Node at the Linköping's University. For the experiments, PBMCs were snap-frozen and thawed according to a SmartTube protocol number STP1FT-120101B. All samples were barcoded using Fluidigm Cell-ID 20-Plex Pd barcoding kit (Fluidigm Inc.) following protocol PN PRD023 V1. Then, $3 \times 10^6$ cells were blocked using human TruStain FcX and purified mouse IgG1 antibody (both from BioLegend) for 5 min, and the appropriate antibody cocktail was added (Table S3). The cells were then incubated on ice for 30 min and washed and fixed in 4% paraformaldehyde (Thermo Fisher Scientific) overnight at 4°C. Next day, the cells were incubated with 0.125 $\mu$M iridium-based DNA intercalator at room temperature for 30 min. They were then washed with a cell-staining buffer (Fluidigm), then with distilled deionized water, and finally resuspended in distilled deionized water supplemented (1:10, vol/vol) with EQ 4 element beads (Fluidigm) at a cell concentration of maximum $5 \times 10^5$ cells/ml. Mass cytometry was performed using CyTOF 2 (Fluidigm) operated as described before (Baumgart et al, 2017). CyTOF 2 instrument was prepared daily for acquisition and tuning, according

to the manufacturer's instructions, using wash and tuning solutions (Fluidigm). The cells were analyzed at a rate of 45 $\mu$l/ml, with noise reduction, in a dual calibration mode and event length set from 10 to 75.

### Cell culture

The human TNBC cell line MDA-MB-231 (ATCC, LGC Standards) was cultured in RPMI-1640 medium supplemented with 1% sodium pyruvate, 1% Hepes, and penicillin/streptomycin mix (100 U/ml and 100 $\mu$g/ml, respectively) (Thermo Fisher Scientific HyClone) and 10% fetal bovine serum (Biosera). Mouse endothelial MS1 cells (ATCC, LGC Standards) were co-cultured with purified CD15$^+$ LDGs or HDGs for 48 h, with or without ROS inhibitor (4000 U/ml catalase; Sigma-Aldrich), and mRNA harvested for analysis.

### Real-time quantitative PCR (RT-qPCR)

RT-qPCR was performed in triplicate using Maxima SYBR Green/Rox (Thermo Fisher Scientific), according to the manufacturer's instructions. The RT-qPCR analysis was performed using the Mx3005P QPCR system (Agilent Technologies). The relative mRNA levels of *Ly6C* and *CX3CL1* were normalized to *GADPH* and *HPRT* levels, and calculated using the comparative Ct method (Vandesompele et al, 2002). Primer sequences are listed in Table S4.

### T-cell suppression assay

The immunosuppressive capacity of sorted G-MDSCs was evaluated by using T-cell suppression assay. Allogeneic donor T cells were enriched by using naïve CD4$^+$ T-cell isolation kit II (Milteny Biotec), according to the manufacturer's instructions. For the analysis, $4 \times 10^4$ freshly isolated naïve CD4$^+$ T cells were seeded into 96-well plates, cultured with CD3/CD28-stimulating Dynabeads (Gibco, Thermo Fisher Scientific), and sorted G-MDSCs at a stimulator-responder ratio of 0 (control), 1:8, 1:4, and 1:2 for 2–7 d. T cells alone were used as a negative control (the base line). The proliferation of responding T cells was quantified by measuring the incorporation of 1 $\mu$Ci [methyl-$^3$H]-thymidine after 2–7 d and an 18-h pulse period. The incorporation was determined using a Microbeta Counter (PerkinElmer). For mechanistic studies, inhibitors of iNOS (100 $\mu$M L-NNA, 99%; Sigma-Aldrich), ARG (50 $\mu$M nor-NOHA; Calbiochem), or ROS (4,000 U/ml catalase; Sigma-Aldrich) were used.

### Animal procedures

Female 8-wk-old NSG mice (NOD.Cg-*Prkdc(scid)Il2rg(tm1Wji)*/SzJ strain; Jackson Laboratory) were housed in a controlled environment. All procedures were approved by the Regional Ethics Committee for Animal Research in Lund/Malmö, Sweden (M11-15), and performed according to the ARRIVE guidelines. $2 \times 10^6$ human TNBC MDA-MB-231 cells were subcutaneously injected on the right flank, alone (n = 4 mice) or in combination with $2 \times 10^5$ primary human G-MDSCs sorted from MBC patient material (n = 4 mice). Tumors were excised on day 21 after injection, fixed in 4% paraformaldehyde, and embedded in paraffin.

## IHC and IF

For IHC and IF analysis, 4-$\mu$m-thick sections of paraffin-embedded tumor were mounted onto glass slides and deparaffinized. This was followed by antigen retrieval using the PT-link system (Agilent) and staining using Autostainer Plus (Agilent) and EnVisionFlex High pH kit (Agilent). Cytospin samples were prepared from sorted G-MDSCs. For the TMA, CD15 was annotated as the absence or presence of infiltration (0, 1) for survival analyses, and as 0–3 for correlation analyses. CD15$^+$MPO$^+$ and CD15$^+$MPO$^-$ cells were annotated as the absence or presence of expression (0, 1). All antibodies and every staining procedure were tested for mouse and human specificity, using mouse spleen or human tonsil sections (the experiments were approved by the Regional Ethical Committees in Lund, Sweden; M149-14 and Dnr 2017/941). The primary antibodies used for IHC and IF are listed in Table S5. Sirius Red staining was performed following clinical pathology routine methods. All histological sections were counterstained with Hematoxylin and Eosin.

## Gene expression analysis

Total RNA was extracted from human monocytes and G-MDSCs from healthy donors and from sepsis and BC patients using Trizol (Invitrogen) according to the manufacturer's instructions. The RNA samples were then treated with DNAse I. RNA integrity was evaluated using Bioanalyzer Pica-kit (Agilent Technologies). Samples were hybridized with Affymetrix Human Gene 1.0 ST array (Illumina Inc.) at the SCIBLU Genomics Center at Lund University (Table S1). The data discussed in this publication have been deposited in National Center for Biotechnology Information (NCBI)'s Gene Expression Omnibus (Edgar et al, 2002) and are accessible through GEO Series accession number GSE157737 (https://www.ncbi.nlm.nih.gov/geo/query/acc.cgi?acc=GSE157737). Quantile normalization was used and low-quality probes (detection $P > 0.01$) were filtered. The RNA yield in one of the HC N samples was too low for further analysis. For gene expression analysis of the Gram-positive sepsis and MBC group, probe sets with a fold change > 2 and $P < 0.05$ between the sepsis and MBC groups were plotted in Heatmap2 in R ([gplots 3.0.3], Rstudio 1.2.1335, R 3.6.3). Hierarchical clustering (method average) was used to compare patient groups. Enhanced Volcano (1.5.0) (Blighe et al, 2019) was used to plot differentially expressed genes (log$_2$ fold-change ± 0.5, $P < 0.05$) between S and MBC groups. Prcomp was used to plot a PCA plot of the gene expression data in R (Rstudio 1.2.1335 and R 3.6.3). In the Volcano and PCA plots, multiple gene names were collapsed to single gene names by choosing the probes with the highest intensity value for each gene in R.

## Angiogenesis pathway analysis

GSEA of the hallmark angiogenesis pathway was performed by extracting the gene expression values from the Affymetrix dataset with the angiogenesis GSEA hallmark gene names matching those of the Affymetrix dataset (Subramanian et al, 2005; Afgan et al, 2016). Multiple gene names were collapsed as above for the Volcano or PCA plots. The resulting values were plotted as a heat map with Heatmap2 in R.

## Statistical analysis

Mass cytometry data were exported as concatenated and randomized FCS 3.0 files using DVS Sciences CyTOF instrument control software v 6.0.626 (DVS Sciences), normalized, and bar de-coded using the stand-alone CyTOF software v 6.7.1014 and normalizing passport EQ-P13H2302_ver2 (Fluidigm). Data were analyzed using ANOVA with multiple comparisons or $t$ test, as indicated in the figure legends. IBM SPSS Statistics v 23.0 (SPSS Inc.), Graph Pad Prism, or R v 3.1.1 software were used for statistical analyses. Correlations to clinicopathological variables were analyzed using Pearson's chi-squared test, Fisher's exact test, or Mann–Whitney U test, as indicated in table legends. The publicly available database R2 (http://r2.amc.nl; microarray analysis and visualization platform) (platform); TCGA 1097 was used for gene expression profile analyses.

## Data and materials availability

All datasets generated in the course of the current study are presented in the main text and the Supplementary Information is available online. The data discussed in this publication have been deposited in NCBI's Gene Expression Omnibus (Edgar et al, 2002) and are accessible through GEO Series accession number GSE157737 (https://www.ncbi.nlm.nih.gov/geo/query/acc.cgi?acc=GSE157737). The TCGA gene expression data were analyzed using publicly available database R2, microarray analysis and visualization platform (platform), and TCGA 1097 was used for gene expression profile analyses.

# Supplementary Information

# Acknowledgements

We thank Mrs Kristina Ekström-Holka for technical support during IHC sample preparation. We also thank National Bioinformatics Infrastructure Sweden (Lund, Sweden) for help with the gene expression profiling and Science for Life Laboratory Node, Linköping, Sweden, for mass cytometry. Elevate Scientific, www.elevatescientific.com, Lund, Sweden, was used for linguistic corrections and improvements. This work was generously supported by grants from the Swedish Research Council (grant number 2017 02443), the Swedish Cancer Society (grant number 18 0693), Funding of Clinical Research within the National Health Service (ALF), Kocks Foundation, Österlunds Foundation, Gunnar Nilsson Cancer Foundation, Malmö Allmänna Sjukhus Cancer Foundation, and Åke Wibergs Foundation and Gyllenstiernska Krapperups foundation.

## Author Contributions

M Mehmeti-Ajradini: conceptualization, data curation, formal analysis, validation, investigation, visualization, methodology, and writing—original draft, review, and editing.
C Bergenfelz: data curation, formal analysis, validation, investigation, visualization, methodology, and writing—review and editing.
A-M Larsson: data curation, formal analysis, validation, investigation, visualization, and writing—review and editing.
R Carlsson: data curation, formal analysis, validation, investigation, visualization, and writing—review and editing.

K Riesbeck: data curation, validation, investigation, writing—review and editing, and responsible for sepsis patient samples.

J Ahl: data curation, validation, investigation, writing—review and editing, and responsible for sepsis patient samples.

H Janols: data curation, formal analysis, validation, investigation, visualization, writing—review and editing, and responsible for sepsis patient samples for flow cytometry.

M Wullt: validation, investigation, visualization, writing—review and editing, and responsible for sepsis patient samples for flow cytometry.

A Bredberg: validation, investigation, visualization, writing—review and editing, and responsible for sepsis patient samples for flow cytometry.

E Källberg: data curation, formal analysis, validation, investigation, visualization, and writing—review and editing.

F Börk Gunnarsdottir: data curation, formal analysis, validation, investigation, visualization, and writing—review and editing.

C Rydberg-Millrud: data curation, formal analysis, validation, investigation, visualization, and writing—review and editing.

L Rydén: data curation, validation, investigation, writing—review and editing, and responsible for breast cancer patient samples for flow cytometry.

G Paul: formal analysis, validation, investigation, visualization, and writing—review and editing.

N Loman: data curation, validation, investigation, visualization, writing—review and editing, and responsible for breast cancer patient samples.

J Adolfsson: data curation, formal analysis, validation, investigation, visualization, and writing—review and editing.

A Carneiro: data curation, validation, investigation, visualization, writing—review and editing, and responsible for melanoma patient samples.

K Jirström: data curation, formal analysis, validation, investigation, visualization, writing—review and editing, and responsible for breast cancer patient TMA.

F Killander: conceptualization, data curation, formal analysis, validation, investigation, visualization, writing—review and editing, and responsible for breast cancer patient samples.

D Bexell: conceptualization, data curation, formal analysis, validation, investigation, visualization, methodology, and writing—review and editing.

K Leandersson: conceptualization, resources, data curation, formal analysis, supervision, funding acquisition, validation, investigation, visualization, methodology, project administration, and writing—original draft, review, and editing.

## Conflict of Interest Statement

K Leandersson is a board member of Cantargia AB, a company developing IL1RAP inhibitors. This does not alter the author's adherence to all guidelines for publication. The authors otherwise declare no competing interest.

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
