## [Reviewer comments · Life Science Alliance]

Life Science Alliance

Human G-MDSCs are Neutrophils at Distinct Maturation Stages Promoting Tumor Growth in Breast Cancer

Meliha Mehmeti-Ajradini, Caroline Bergenfelz, Anna-Maria Larsson, Robert Carlsson, Kristian Riesbeck, Jonas Ahl, Helena Janols, Marlene Wullt, Anders Bredberg, Eva Källberg, Frida Björk Gunnarsdottir, Camilla Rydberg-Millrud, Lisa Rydén, Gesine Paul, Niklas Loman, Jörgen Adolfsson, Ana Carneiro, Karin Jirström, Fredrika Killander, Daniel Bexell, and Karin Leandersson

DOI: <https://doi.org/10.26508/lsa.202000893>

Corresponding author(s): Karin Leandersson, Lund University

Review Timeline:	Submission Date:	2020-08-25
	Editorial Decision:	2020-08-25
	Revision Received:	2020-09-08
	Editorial Decision:	2020-09-08
	Revision Received:	2020-09-10
	Accepted:	2020-09-10

Scientific Editor: Shachi Bhatt

Transaction Report:

Please note that the manuscript was previously reviewed at another journal and the reports were taken into account in the decision-making process at Life Science Alliance. Since the original reviews are not subject to Life Science Alliance's transparent review process policy, the reports and author response cannot be published.

August 25, 2020

Re: Life Science Alliance manuscript #LSA-2020-00893-T

Dr. Karin Leandersson
Lund University
Department of Translational Medicine
Center for Molecular Pathology, Entr 78 SUS-M, Lund University
Malm? 20502

Dear Dr. Leandersson,

Thank you for transferring your manuscript entitled "Human G-MDSCs are Neutrophils at Distinct Maturation Stages that Promote Tumor Growth and Immune Cell Exclusion in Breast Cancer" to Life Science Alliance (LSA).

FOR A BRIEF OVERVIEW: The manuscript was reviewed at another journal, and transferred to LSA with the reviewers' comments by the authors and with the help of editors of the previous journal. The reviewers had expressed concerns about the low sample size and interpatient variability. They did not think that the conclusions drawn could be supported without rigorously addressing those 2 concerns. At LSA, we agreed that if the authors tone down their conclusions (in accordance with the data presented), acknowledge the sample size concerns, and discuss the inter-patient variability, the findings are still sufficiently valuable for the community to merit publication.

Based on the reviewers comments from the previous journal the manuscript should be acceptable for publication at LSA with the following revisions

1. please tone down the conclusions, based on the critiques of both reviewers at the previous journal
2. please address Rev 1's pt 3 and 6 experimentally - if possible, this would strengthen study, however would not be required for publication at Life Science Alliance
3. please provide a discussion on the potential inter-patient heterogeneity that might affect these conclusions
4. Along with the revised manuscript, please also provide a pbp rebuttal to the reviewers' comments from the previous journal, and a marked up manuscript file that highlights all the changes made.

We would be happy to discuss the individual revision points further with you should this be helpful. The typical timeframe for revisions is three months. Please note that papers are generally considered through only one revision cycle, so strong support from the referees on the revised version is needed for acceptance. When submitting the revision, please include a letter addressing the reviewers' comments point by point.

Please use the link below to submit your revised manuscript
<https://lsa.msubmit.net/cgi-bin/main.plex>

Thank you for considering Life Science Alliance (LSA) as an appropriate venue for your research. Please email me with any questions.

Sincerely,

Shachi Bhatt
Executive Editor
Life Science Alliance

B. MANUSCRIPT ORGANIZATION AND FORMATTING:

September 8, 2020

RE: Life Science Alliance Manuscript #LSA-2020-00893-TR

Prof. Karin Leandersson
Lund University
Department of Translational Medicine
Cancer Immunology, CRC
Malmö 21428
Sweden

Dear Karin,

Thank you for submitting your revised manuscript entitled "Human G-MDSCs are Neutrophils at Distinct Maturation Stages Promoting Tumor Growth in Breast Cancer". We would be happy to publish your paper in Life Science Alliance pending final revisions necessary to meet our formatting guidelines.

Along with the points listed below, please also address the following in the revised manuscript,

- please use the [10 author names, et al.] format in your references (i.e. limit the author names to the first 10)
- please upload your supplementary figures as single files
- please include a scale bar for the PDGFR β sections in Fig S3
- please deposit the microarray data in the NCBI database, and include the accession number in the revised manuscript text

A. FINAL FILES:

B. MANUSCRIPT ORGANIZATION AND FORMATTING:

Sincerely,

Shachi Bhatt, Ph.D.
Executive Editor
Life Science Alliance

September 10, 2020

RE: Life Science Alliance Manuscript #LSA-2020-00893-TRR

Prof. Karin Leandersson
Lund University
Department of Translational Medicine
Cancer Immunology, CRC
Malmö 21428
Sweden

Dear Dr. Leandersson,

Thank you for submitting your Research Article entitled "Human G-MDSCs are Neutrophils at Distinct Maturation Stages Promoting Tumor Growth in Breast Cancer". It is a pleasure to let you know that your manuscript is now accepted for publication in Life Science Alliance. Congratulations on this interesting work.

DISTRIBUTION OF MATERIALS:

Again, congratulations on a very nice paper. I hope you found the process to be constructive and are pleased with how the manuscript was handled editorially. We look forward to future exciting submissions from your lab.

Sincerely,

Shachi Bhatt, Ph.D.
Executive Editor
Life Science Alliance